# In situ cryo-ET structure of phycobilisome–photosystem II supercomplex from red alga

Meijing Li[1†‡], Jianfei Ma[2†], Xueming Li[1]*, Sen-Fang Sui[2,3]*

[1]Key Laboratory for Protein Sciences of Ministry of Education, Beijing Advanced Innovation Center for Structural Biology & Frontier Research Center for Biological Structure, School of Life Sciences, Tsinghua University, Beijing, China; [2]State Key Laboratory of Membrane Biology, Beijing Advanced Innovation Center for Structural Biology & Frontier Research Center for Biological Structure, School of Life Sciences, Tsinghua University, Beijing, China; [3]Department of Biology, Southern University of Science and Technology, Guangdong, China

**\*For correspondence:**
lixueming@mail.tsinghua.edu.
cn (XL);
suisf@mail.tsinghua.edu.cn (S-
FangS)

[†]These authors contributed
equally to this work

**Present address:** [‡]Max Planck
Institute of Biochemistry,
Department of Molecular
Structural Biology, Martinsried,
Germany

**Competing interest:** The authors
declare that no competing
interests exist.

**Reviewing Editor:** Andrew
P Carter, MRC Laboratory of
Molecular Biology, United
Kingdom

**Abstract** Phycobilisome (PBS) is the main light-harvesting antenna in cyanobacteria and red algae. How PBS transfers the light energy to photosystem II (PSII) remains to be elucidated. Here we report the in situ structure of the PBS–PSII supercomplex from *Porphyridium purpureum* UTEX 2757 using cryo-electron tomography and subtomogram averaging. Our work reveals the organized network of hemiellipsoidal PBS with PSII on the thylakoid membrane in the native cellular environment. In the PBS–PSII supercomplex, each PBS interacts with six PSII monomers, of which four directly bind to the PBS, and two bind indirectly. Additional three 'connector' proteins also contribute to the connections between PBS and PSIIs. Two PsbO subunits from adjacent PSII dimers bind with each other, which may promote stabilization of the PBS–PSII supercomplex. By analyzing the interaction interface between PBS and PSII, we reveal that $\alpha^{LCM}$ and ApcD connect with CP43 of PSII monomer and that $\alpha^{LCM}$ also interacts with CP47' of the neighboring PSII monomer, suggesting the multiple light energy delivery pathways. The in situ structures illustrate the coupling pattern of PBS and PSII and the arrangement of the PBS–PSII supercomplex on the thylakoid, providing the near-native 3D structural information of the various energy transfer from PBS to PSII.

## Introduction

Life on Earth depends on photosynthesis for the conversion of solar energy to chemical energy. Red algae living in deep water can efficiently use dim and green light that cannot be absorbed by plants (*Tschudy, 1934*). This ability is mainly due to two primary photosynthetic components: the light-harvesting antenna and the photochemical reaction centers. The primary light-harvesting antenna is the phycobilisome (PBS), which is located on the stromal surface of the thylakoid membrane and captures light energy (*Gantt and Conti, 1965*; *Gantt and Conti, 1966*). The two reaction centers, photosystem II (PSII) and photosystem I (PSI), are located in the thylakoid membrane. PBS mainly transfers the light energy to PSII, leading to water splitting (*Biggins and Bruce, 1989*; *Ley and Butler, 1977a*). PBS could also directly couple with PSI (*Mullineaux, 1992*; *Su et al., 1992*; *Kondo et al., 2007*; *Ueno et al., 2017*).

PBSs are composed of chromophore-bearing phycobiliproteins (PBPs) and linker proteins, which further assemble into central core and peripheral rods (*Gantt et al., 1976*; *Zilinskas and Greenwald, 1986*). Recently, the cryo-electron microscope (cryo-EM) structures of PBS from red algae at resolutions of 3.5 Å (*Zhang et al., 2017*) and 2.82 Å (*Ma et al., 2020*). The cryo-EM structure of PBS in red

alga *Porphyridium purpureum* (*Ma et al., 2020*) shows that the PBS consists of a tricylindrical PBS core, 14 rods (Rod a–Rod g, Rod a'–Rod g'), 8 individual extra PE $(\alpha\beta)_6$ hexamers (Ha–Hd, Ha'–Hd'), and 24 individual extra PE α or β subunits. The core contains one top cylinder B, composed of two allophycocyanin (APC) $(\alpha\beta)_3$ trimers, and two bottom cylinders (A and A'), each of which is assembled by three APC trimers (A1–A3, A2 and A3 form APC hexamer). The rods consist of phycoerythrin (PE) and phycocyanin (PC) hexamers or only PE hexamers. For example, Rod a is composed of one basal PC hexamer and two distal PE hexamers. Thus, the energy absorbed by Rod transfers unidirectionally from the distal PE to the basal PC, and then funnels to APC in the core, and eventually to the two terminal emitters, including chromophores in the core–membrane linker protein ($L_{CM}$, also called ApcE) (*Capuano et al., 1991*; *Lundell et al., 1981*; *Tang et al., 2015*) and allophycocyanin D (ApcD) (*Glazer and Bryant, 1975*; *Peng et al., 2014*).

PSII is a multi-pigment transmembrane protein complex involved in converting light energy into electrochemical potential energy (*Adachi et al., 2008*). The X-ray crystal structures of PSII from red alga *Cyanidium caldarium* (*Ago et al., 2016*), cyanobacterium *Thermosynechococcus elongatus* (*Nakajima et al., 2018*), and the cryo-EM structure of PSII from green alga *Chlamydomonas reihardtii* (*Sheng et al., 2019*), the marine diatom *Chaetoceros gracilis* (*Pi et al., 2019*), and the higher plants *Arabidopsis thaliana* (*van Bezouwen et al., 2017*), *Spinacia oleracea* (*Wei et al., 2016*), and *Pisum sativum* (*Su et al., 2017*) have revealed the structural basis of energy transfer, electron transfer, and photoprotection within the photosystem. The main subunits of PSII include the reaction center D1, D2 protein, and chlorophyll-a binding proteins, CP43 and CP47. Both CP43 and CP47 subunits have been reported to mediate the energy transfer from PBS to the reaction center (*Ueno et al., 2017*).

The structural mechanism of energy transfer from PBS to PSII has been studied for many years. The organization of PBS and PSII on the thylakoid membrane has been studied using ultrathin section (*Gantt and Conti, 1965*; *Wanner and Kost, 1980*), freeze-fracture analysis (*Lange et al., 1990*), negative staining (*Arteni et al., 2008*; *Chang et al., 2015*; *Folea et al., 2008*; *Hellmich et al., 2014*), atomic force microscopy (*Liu et al., 2008*), and cryo-electron tomography (cryo-ET) (*Levitan et al., 2019*; *Rast et al., 2019*; *Wietrzynski et al., 2020*). PBSs in red algae are tightly arranged in ordered rows on the thylakoid in most regions and randomly packed in some areas (*Arteni et al., 2008*; *Lange et al., 1990*; *Liu et al., 2008*). PBS arrangement in the two regions is dynamically regulated, varying with the cell's age, the supply of nutrients, and available light (*Arteni et al., 2008*; *Liu et al., 2008*; *Sagert and Schubert, 1995*). However, the stoichiometric ratio of PBS/PSII in red algae and cyanobacteria remains controversial, ranging from 1:1 to 1:4 (*Arteni et al., 2008*; *Cunningham et al., 1989*; *Ohki et al., 1987*; *Takahashi et al., 2009*), which likely reflects different growth conditions. Different approaches showed that PBS connects with PSII through ApcD, $\alpha^{LCM}$ (the α domain of $L_{CM}$), and ApcF (*Zhao et al., 1992*). In particular, ApcD and $\alpha^{LCM}$ are well known as terminal emitters for mediating the transfer of energy from PBS to PSII. The negative staining structure of the purified PBS–PSII supercomplex in the *Anabaena* sp. strain PCC 7120 proposed that $\alpha^{LCM}$ and ApcF of PBS play important roles in mediating PBS interaction with PSII (*Chang et al., 2015*). The cross-linking structure of the purified PBS–PSII–PSI megacomplex in the cyanobacteria *Synechocystis* PCC 6803 identified five interlinks associated with $\alpha^{LCM}$ and PSII (*Liu et al., 2013*). However, the organization and precise connection between PBS and PSII in the cellular environment are not fully understood.

In situ cryo-ET in combination with cryo-focused ion beam (cryo-FIB) milling is currently a powerful approach for directly dissecting macromolecular structures in the unperturbed cellular environment at molecular resolution (*Wagner et al., 2020*; *Watanabe et al., 2020*). The obtained structure is complementary to the structure of the isolated complex and reveals elements of the in vivo organization that can never be obtained through structures of the isolated complex, no matter how high the resolution. Here, by using this approach, we identified the native PBS–PSII supercomplex and the double PBS–PSII supercomplex containing two adjacent PBS–PSII supercomplexes at resolutions of 14.3 Å and 15.6 Å, respectively. These results illustrate the coupling pattern of PBS and PSII and the arrangement of PBS–PSII supercomplexes on the ordered distribution region of the thylakoid membrane, which cannot be determined with other methods. The newly found structural information provides a better understanding of energy transfer from PBS to PSII.

## Results

### Visualization of PBS and PSII on the thylakoid membrane

The red alga, *P. purpureum*, was cultured under low-light conditions to mimic its natural habitat (*Cunningham et al., 1989*) and to increase the population of regularly distributed PBS–PSII supercomplexes (*Liu et al., 2008*). Cells were harvested in the exponential phase of growth and were immediately vitrified on cryo-EM grids to minimize environmental affection (*Kaňa et al., 2014*). Cryo-FIB milling was then used to prepare thin lamellae with ~150 nm thickness for cryo-ET data collection. We first imaged the whole cells. The tomogram slice shows that most PBS–PSII supercomplexes are packed in an orderly fashion on the thylakoid membranes (*Figure 1—figure supplement 1A*). Subsequently, 51 cryo-ET tilt series were collected at higher magnification at the ordered distribution regions for three-dimensional (3D) reconstruction and further sub-tomogram averaging.

Consistent with the conventional transmission electron microscope ultrastructures of other red algae (*Gantt and Conti, 1965*; *Gantt et al., 1968*; *Tsekos et al., 1996*; *Wanner and Kost, 1980*), PBSs are sandwiched between parallel thylakoid membranes. In the cross-section view, adjacent PBSs were attached to the upper and lower thylakoid membrane in opposite directions (*Figure 1A,B,D*). In the two orthogonal views, opposite PBSs are stacked into linear rows respectively (*Figure 1C,E*, *Figure 1—figure supplement 1B,C*, *Video 1*, *Video 2*, *Video 3*). These PBS rows are arranged in parallel, forming a large two-dimensional pattern covering the thylakoid membrane. In the cross-section view, densities are evident at the bottom of PBSs, which are embedded within the thylakoid membrane. Protrusions are visible on the luminal side (*Figure 1A*). The features of these densities are consistent with the location and morphology of PSIIs (*Ago et al., 2016*; *Chang et al., 2015*; *Lange et al., 1990*). From this, we can infer that these represent PSIIs. Due to the small size of the extrinsic domain of PSI, we could not distinguish convincing PSI densities in our tomograms.

To verify the PSII densities, we performed sub-tomogram averaging of two adjacent PBS particles on the same linear array (indicated in the red boxes of *Figure 1—figure supplement 1B,C*) and generated a structure at a resolution of 15.6 Å (*Figure 2A*, *Figure 2—figure supplement 1A,B*). The recently determined 2.82 Å single-particle PBS (PDB code 6 KGX) (*Ma et al., 2020*) was well-fitted into our sub-tomogram map (*Supplementary file 1*), except for the individual extra PE β subunit between the Rod e and Rod d' (*Figure 2—figure supplement 2A,B*). The map's slice view shows a similar PSII dimer array structure resembling that of the red algae *Porphyridium cruentum* (*Lange et al., 1990*; *Figure 2—figure supplement 3A,B*).

To find the most likely PSII atomic resolution structure to fit the subtomogram map, we analyzed all of the reported X-ray models in a phylogenetic analysis (*Figure 2—figure supplement 4B*). PSII from the red algae *C. caldarium* (PDB code 4YUU) (*Ago et al., 2016*) is, evolutionarily, most similar to *P. purpureum*, and almost all the subunits show high homogeneity (ranging from 35% to 100%) with that of *P. purpureum* (*Figure 2—figure supplement 4A,C*). We then docked the X-ray model of PSII from *C. caldarium* into the sub-tomogram map, which shows a high cross-correlation coefficient (all above 80%) between the model and sub-tomogram map (*Supplementary file 1*). Thus, we further confirmed these densities as PSIIs.

### The overall structure of the PBS–PSII supercomplex

Our sub-tomogram map also reveals a different stoichiometry of PBS and PSII from previous reports (*Arteni et al., 2008*; *Cunningham et al., 1989*; *Ohki et al., 1987*; *Takahashi et al., 2009*). In the sub-tomogram map, two PBSs (labeled as PBS1 and PBS2) interact with a linear array of PSII dimers (labeled as A, B, C, A', B', C', C", *Figure 2A* (left panel), *Figure 2—figure supplement 3B*). Considering the map contains two PBSs, we named the structure double PBS–PSII supercomplex. After mapping this sub-tomogram map back into the tomogram, we observed long linear arrays of the PBS–PSII supercomplexes on the thylakoid membrane (*Figure 1B–E*). In this array, the PBS/PSII ratio is 1:6, that is, one PBS with six associated monomeric PSIIs periodically repeated along with the linear array. In the double PBS–PSII map, each PBS is directly anchored to a pair of PSIIs dimers, i.e., PBS1 on the A–B pair and PBS2 on the A'–B' pair (*Figure 2A*, right panel). The C' dimer connects the A–B pair to the A'–B' pair, while the two PSIIs in the C' dimer respectively interact with the lateral disc-like densities on both sides of PBS, which have features of PBS hexamer (discussed later), and thus indirectly attach to PBS1 and PBS2 (*Figure 2A*). The slice view shown in *Figure 2—figure supplement 3B* indicates that the PSII dimers are arranged repeatedly in parallel. Thus, C and C" should also be

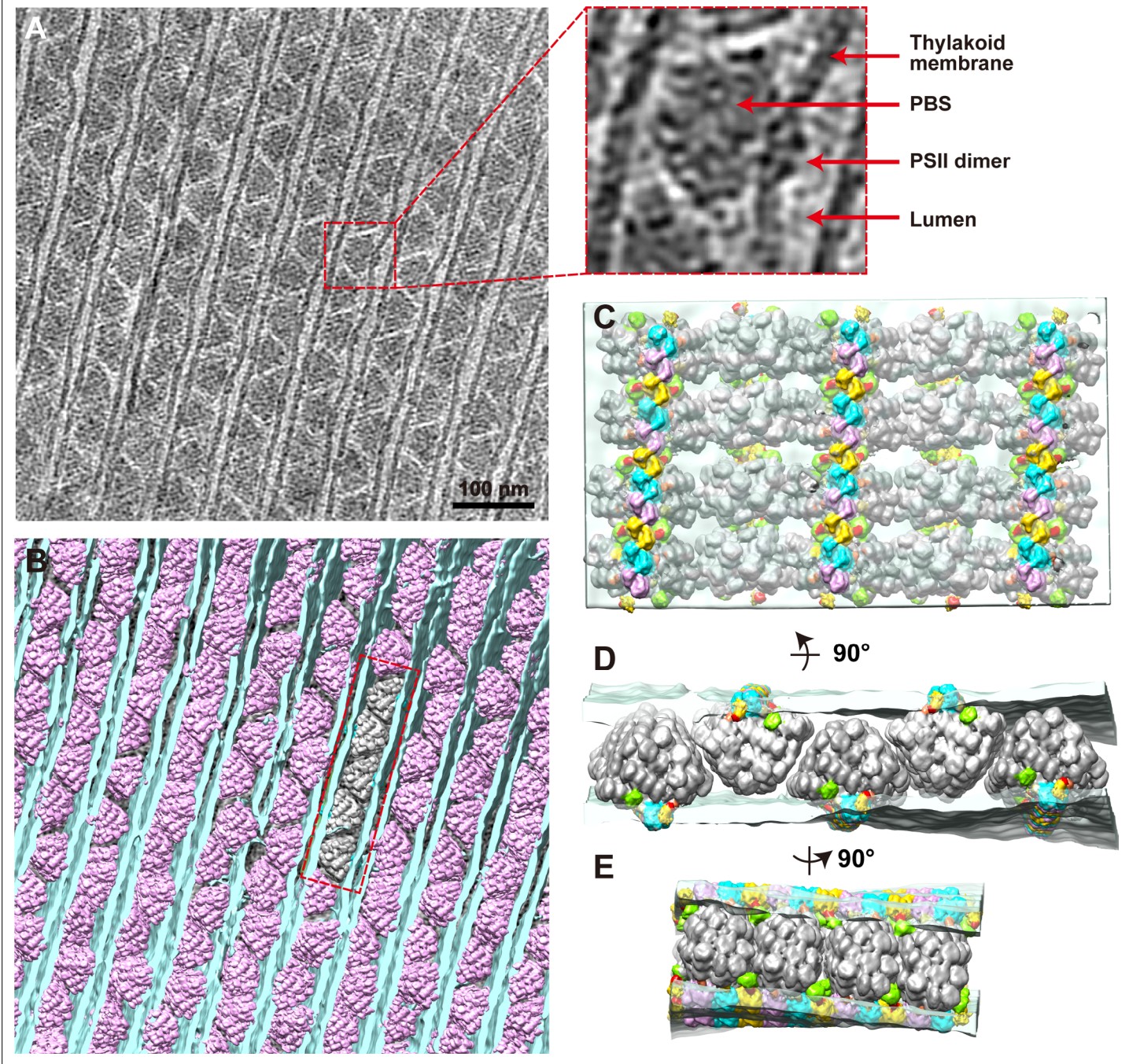

**Figure 1.** Organization of PBS–PSII supercomplexes on the thylakoid membrane. (**A**) Representative tomogram slice in cross-section view. The box represents magnified details of PBS, a PSII dimer, the thylakoid membrane, and the lumen. (**B**) Spatial mapping of the PBS–PSII supercomplex (purple) and 3D segmentation of the thylakoid membrane (blue). (**C–E**) The magnified three presents perpendicular views of the organization of PBS–PSII in the thylakoid membrane boxed in (**B**). Thylakoid membrane, light blue; PBS, gray; Lateral hexamer, green; A1–A2 pair, purple; B1–B2 pair, cyan; C1–C2 pair, yellow; connector 1, coral; connector 2, khaki; connector 3, red.

The online version of this article includes the following figure supplement(s) for figure 1:

**Figure supplement 1.** Tomographic slice of the whole cell and the PBS–PSII supercomplexes on the thylakoid membrane.

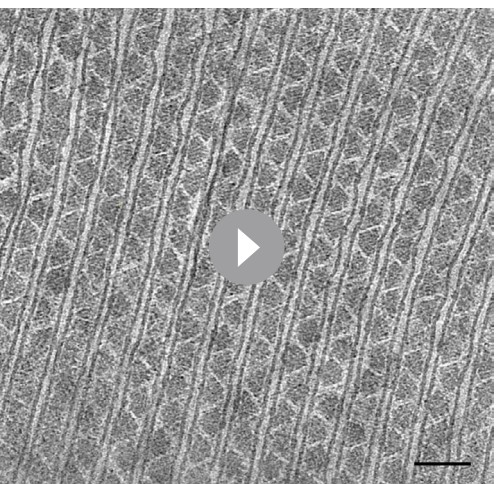

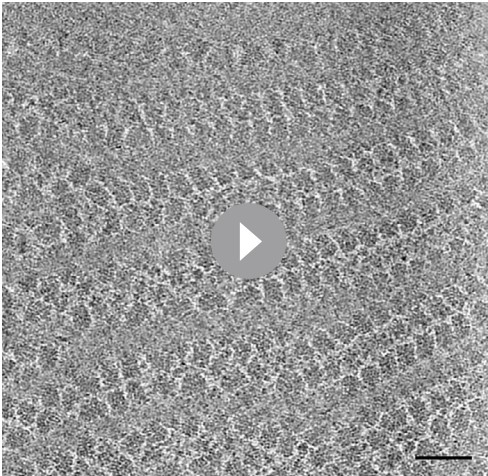

**Video 1.** Sequential slices back and forth through the representative tomogram in cross-section view. Related to Figure 1. Scale bar, 100 nm.
https://elifesciences.org/articles/69635/figures#video1

**Video 3.** Sequential slices back and forth through the representative tomogram slice in cross-section view. Related to Figure 1—figure supplement 1C. Scale bar, 100 nm.
https://elifesciences.org/articles/69635/figures#video3

the PSII dimer, although their densities are weak. The structure shows PBS1 is associated with PSII monomer A1, A2, B1, B2, C1, and C'2, and PBS2 connect with A'1, A'2, B'1, B'2, C'1, C"2. Therefore, each PBS is associated with six monomeric PSIIs.

To study the geometry of the PBS–PSII supercomplex, we manually picked 34,380 individual PBSs with the interacting PSIIs particles. These were aligned and averaged into a structure at a higher resolution, 14.3 Å (*Figure 2B*, *Figure 2—figure supplement 1A,C*). In the PBS–PSII map, the PSII dimers (PSII dimers A and B) are clearly shown. With a large threshold, the other two PSIIs corresponding to PSII dimer C in double PBS–PSII are visualized in the map and tomogram slice view (indicated with red stars in *Figure 2—figure supplement 3C,D*). The distance between the PSII dimers A and B is approximately 12.4 nm, and the face planes of the two PSII dimers make a lateral angle of approximately 14° with the plane of the PBS central core. The centers of the two PSII double dimers shift approximately 3 nm along the plane of the PBS central core, which makes the twofold symmetry axes of PBS and 6 PSIIs coincide (*Figure 2—figure supplement 5*).

The improved PBS–PSII structure enables us to confirm that PSII dimers A and B bind with each other through a rigid density. By analyzing the fitting results, we suppose the density is most likely contributed by the two adjacent PsbO subunits: one is from the PSII monomer 1 of the PSII dimer A and the other from the PSII monomer 4 of the PSII dimer B (*Figure 2*, *Figure 2—figure supplement 2C–E*). However, we are limited by the current resolution and could not exclude that other factors may contribute to the density. The PsbO protein is found in PSII of all oxygenic organisms ranging from cyanobacteria and red algae to higher plants and serves as the manganese-stabilizing protein directly involved in the stability of photosynthetic water-oxidizing complex (*Ifuku and Noguchi, 2016*; *Popelkova*

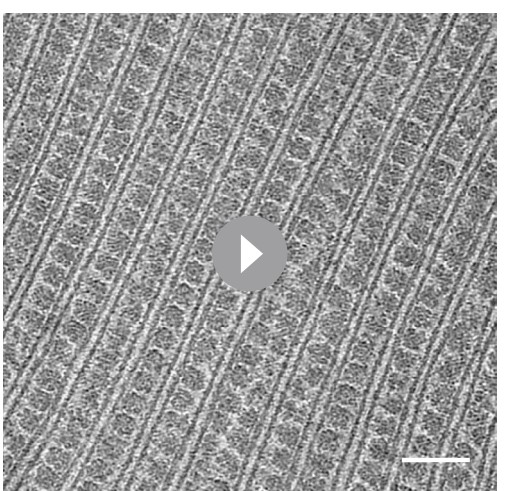

**Video 2.** Sequential slices back and forth through the representative tomogram slice in cross-section view. Related to Figure 1—figure supplement 1B. The color boxes indicate the two types of the tight perforations. Scale bar, 100 nm.
https://elifesciences.org/articles/69635/figures#video2

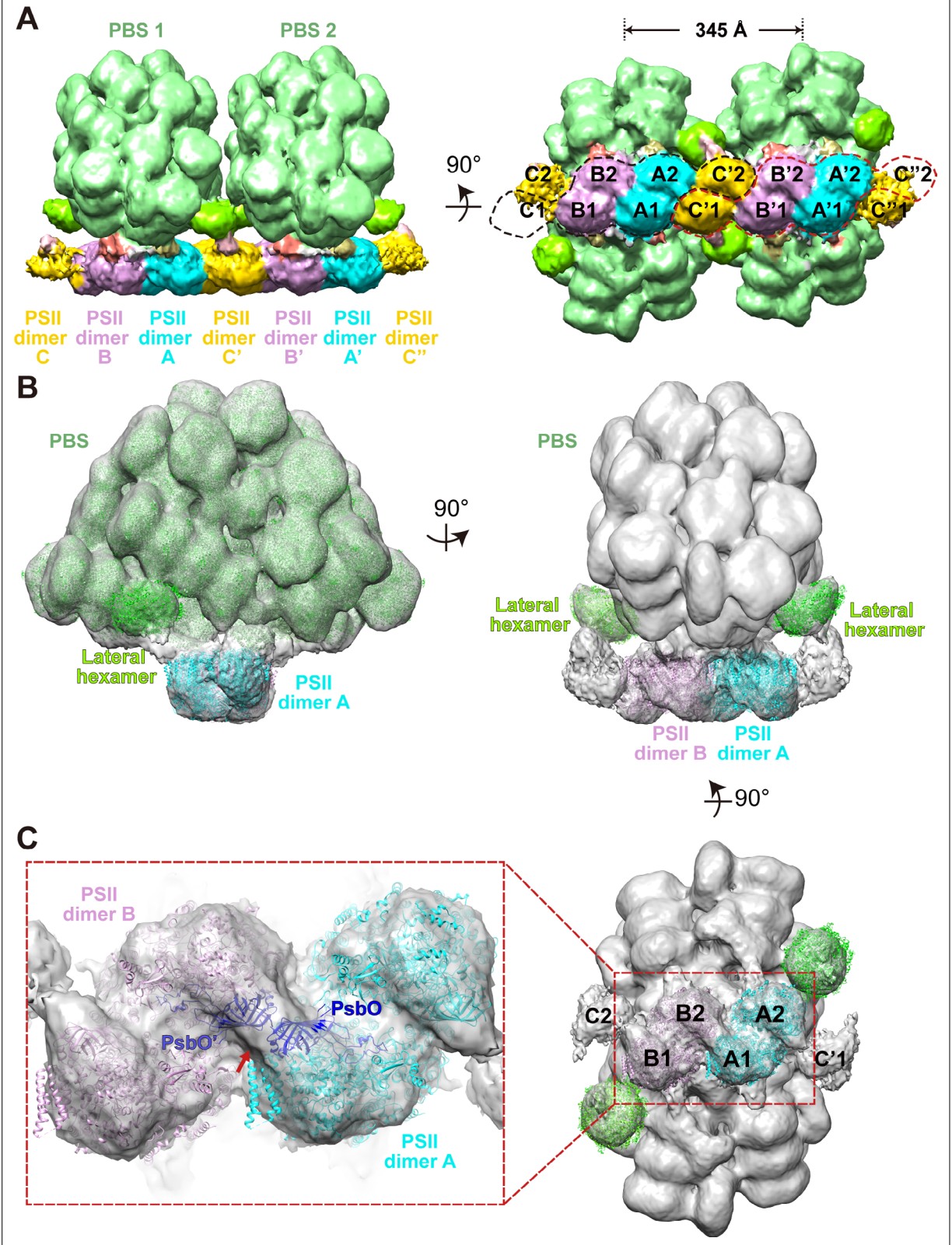

**Figure 2.** Overview of the PBS–PSII and double PBS–PSII structures. (**A**) The density map of double PBS–PSII structures at a resolution of 15.6 Å, presented in two perpendicular views. The center-to-center distance of two adjacent PBSs is approximately 345 Å. Two random circles indicated by black or red dashed lines mark the six PSII monomers binding with each of the two PBSs. PBS1 is associated with PSII monomer A1, A2. B1, B2, C1, and C'2. PBS2 connects with A'1, A'2, B'1, B'2, C'2, and C"1. The surface threshold is 0.09. (**B**) The density map of the PBS–PSII supercomplex at a resolution

*Figure 2 continued on next page*

*Figure 2 continued*

of 14.3 Å fitted with the single-particle model of PBS (EMDB code EMD-9976, PDB code 6 KGX) and X-ray structure of PSII (PDB code 4YUU), presented in two perpendicular views. The lateral hexamer was fitted with the single-particle model of the Rod a distal PE hexamer. The surface threshold is 0.059. (**C**) The magnified image shows that the two PsbO subunits bind with each other at the interface of the adjacent PSII dimers (Inset). A map of PSII dimers A and B, segmented from (**B**) with the same surface threshold level. The arrow indicates the binding site.

The online version of this article includes the following figure supplement(s) for figure 2:

**Figure supplement 1.** Resolution estimation and the ResMap analysis of the density maps for the double PBS–PSII supercomplex and the PBS–PSII supercomplex.

**Figure supplement 2.** Sub-tomogram structural analysis of the PBS–PSII supercomplex.

**Figure supplement 3.** Sub-tomogram map analysis of the double PBS–PSII supercomplex, the PBS–PSII supercomplex, and the PBS–PSII supercomplex with post-processing.

**Figure supplement 4.** PSII sequence analysis of *Porphyridium purpureum* and *Cyanidium caldarium*.

**Figure supplement 5.** Geometry of PBS and PSIIs.

*and Yocum, 2011*). Our results suggest that PsbO could play an additional role in stabilizing adjacent PSII dimers in PSII dimer rows.

To analyze the lateral densities on both sides of PBS connecting PBS to PSII dimer C, we further improved the resolution of the PBS–PSII supercomplex to 13.2 Å after post-processing from the 14.3 Å resolution map (*Figure 2—figure supplement 1D*). The slice view of the PBS–PSII map shows that the densities are round discs with solid densities in the center (*Figure 2—figure supplement 3E–G*). These features are reminiscent of the structure of the extra hexamer found in the single-particle cryo-EM structure of *P. purpureum* PBS. Moreover, the lateral densities are of the same size and shape as the extra hexamer. As all eight extra hexamers in *P. purpureum* PBS are PE hexamers, we hypothesize that the lateral density could be contributed by a PE hexamer, which is referred to as the lateral hexamer (*Figure 2B*). Further analysis indicated that the lateral hexamer connects with the bottom hexamer of Rod a and the second hexamer of Rod e, which are the PC and PE hexamers, respectively (*Figure 2—figure supplement 3E*).

## Interaction pattern between PBS and PSII

In the PBS–PSII map, extensive interfaces between PBS core and PSII are observed. To better analyze the potential interactions, we docked the atomic models of PBS and PSII into the map and extracted the densities of PBS core cylinders A, A' and PSII dimers A, B (*Figure 3A,B*; *Figure 3—figure supplement 1*). ApcD and $\alpha^{LCM}$ are two well-documented terminal emitter subunits of PBS (*Zhao et al., 1992*; *Zlenko et al., 2019*). We observed the interfaces between $\alpha^{LCM}$, ApcD, and PSII dimers A, B from the surface clip views (*Figure 3*, *Video 4*).

As was shown in the proximal and middle clip views, there is a wide interface between $\alpha^{LCM}$ and CP43 of PSII dimer A1 (CP43$^{A1}$), as well as a relatively small interface between $\alpha^{LCM}$ and CP47 of PSII dimer B1 (CP47$^{B1}$) (*Figure 3C–E*). These observations suggested the interactions between $\alpha^{LCM}$ with CP43 and CP47, consistent with the energy transfer from PBS to CP43 and CP47 detected in red algae and cyanobacteria (*Ueno et al., 2017*). The different sizes of the interface may provide a structural view that energy transfer from PBS to CP43 is the main pathway in red alga (*Ueno et al., 2017*). The other terminal emitter subunit ApcD was also involved in energy transfer from PBS to PSII (*Ashby and Mullineaux, 1999*; *Calzadilla et al., 2019*; *Ley and Butler, 1977b*). The middle and distal clip views showed an interaction interface between ApcD and CP43$^{A1}$ (*Figure 3E,F*), which may mediate the energy transfer from ApcD to PSII.

## Supplementary interactions intermediated by three connector proteins

After docking the atomic models of PBS, PSII dimer, and two lateral hexamers into the density map of PBS–PSII supercomplex, we still observed three extra densities that are not occupied by any model. Two of them are associated with the PBS and PSII (*Figure 4—figure supplement 1A*), and the third one is associated with the lateral hexamer and PSII (*Figure 4—figure supplement 1B*). Since we could not identify the proteins corresponding to these densities, we temporarily deemed them 'connectors' (connectors 1–3; *Figure 4*, *Figure 4—figure supplement 2A–E*). Nevertheless, their positions

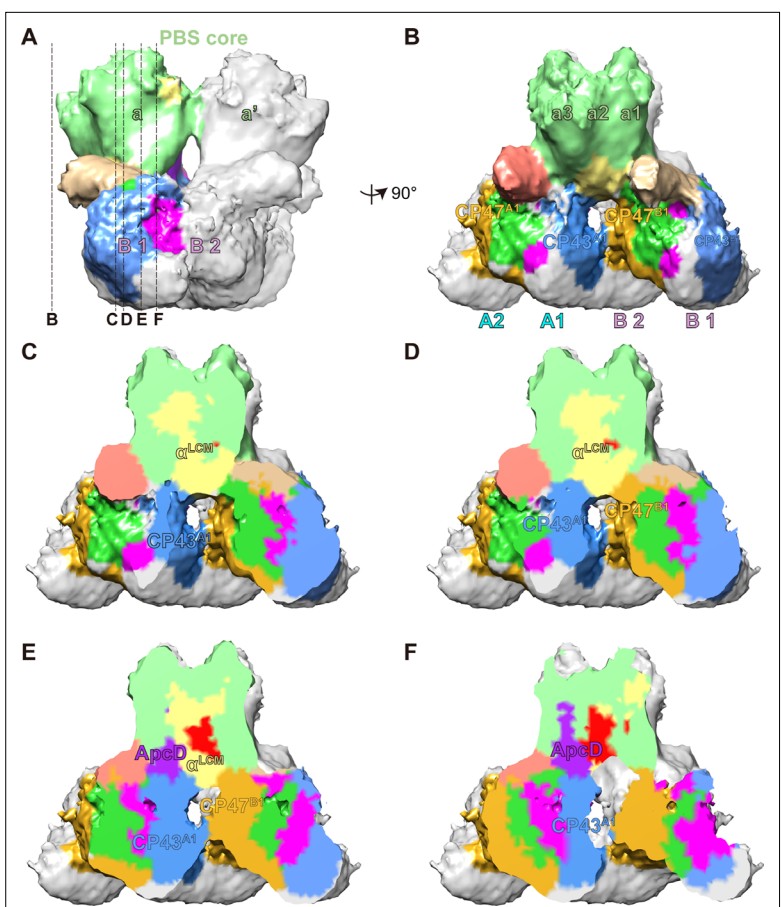

**Figure 3.** The connections between PBS and PSIIs. (**A, B**) A segmentation map of two basal cylinders of the PBS core (labeled as a and a', respectively) and PSII dimers A and B (each monomer is labeled as A1, A2, B1, and B2, respectively). Each basal cylinder consists of three APC trimers, a1, a2, and a3. This map was segmented from the PBS–PSII supercomplex at a surface threshold level of 0.065. (**C–F**) Surface clip views of the segmented PBS–PSII supercomplex show the details of the interaction between the $\alpha^{LCM}$, ApcD, and CP43, CP47. The clip planes were indicated in (**A**). The surface threshold was 0.065. CP43$^{A1}$, CP43 of PSII dimer A1; CP47$^{B1}$, CP47 of PSII dimer B1.

The online version of this article includes the following figure supplement(s) for figure 3:

**Figure supplement 1.** Map extraction model of the PBS–PSII supercomplex.

in the map suggest that these proteins likely participate in the formation of the PBS–PSII super-complex (*Figure 4*) as well as the assembling of the PBS–PSII array on the thylakoid membrane (*Figure 1C–E*).

We segmented out the three connectors from the PBS–PSII supercomplex map and find that they are all rod-like structures, but different in size (*Figure 4—figure supplement 2*). The surface clip views of connector 1 show that connector 1 contacts the β2 subunit of the PBS core layer a3 and the CP47 of PSII monomer B2 (CP47$^{B2}$), as well as D1$^{B2}$ and D2$^{B2}$ (*Figure 4B,C*). Connector 2 is smaller than connector 1 (*Figure 4D,E*, *Figure 4—figure supplement 2A–C*). One side of connector 2 interacts with the basal hexamer of Rod a, and the other side with D1$^{B1}$, D2$^{B1}$, and CP47$^{B1}$ in the neighboring PSII monomer B1

**Video 4.** Clip views of the segmentation map of two basal cylinders of the PBS. Related to Figure 3. https://elifesciences.org/articles/69635/figures#video4

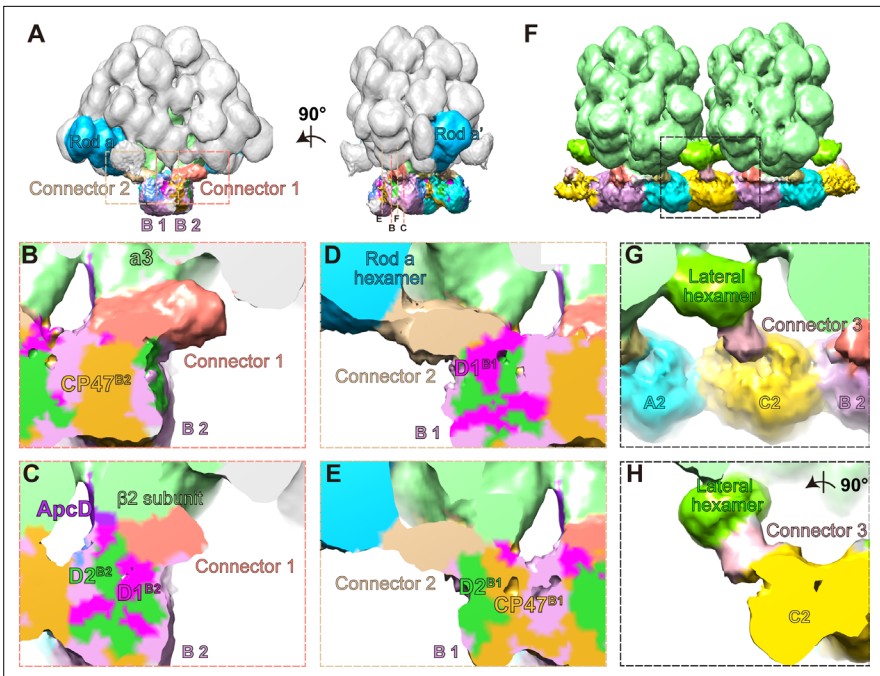

**Figure 4.** Three connector proteins in the PBS–PSII supercomplex. (**A**) Overall structure of the PBS–PSII supercomplex (surface threshold of 0.065), highlighting the PBS core, Rod a of PBS, connector 1, and connector 2 between PBS and PSII. (**B, C**) The magnified clip views of connector 1 show that connector 1 interacts with the β2 subunit of the a3 trimer layer and the CP47, D1, and D2 subunits of PSII monomer B2 (labeled with CP47[B2], D1[B2], and D2[B2], respectively). (**D, E**) The magnified clip views of connector 2 show that connector 2 mediates the connections between the bottom hexamer of Rod a and D1, D2, and CP47 of PSII monomer B1 (labeled with D1[B1], D2[B1], and CP47[B1], respectively). (**F**) The overall structure of double PBS–PSII supercomplex (surface threshold = 0.09) highlights the linkage of connector 3 between PBS's lateral hexamer and PSII's bridging dimer C shown in *Figure 2A*. (**G, H**) The density of connector 3 and its putative connections with PBS's lateral hexamer and PSII from two perpendicular views. According to the slice view of the PBS–PSII supercomplex, some densities of connector 3 are inserted into the central cavity of the lateral hexamer (*Figure 4—figure supplement 2E*); this cavity density is not colored here.

The online version of this article includes the following figure supplement(s) for figure 4:

**Figure supplement 1.** Extra density analysis of the PBS–PSII supercomplex and the double PBS–PSII supercomplex.

**Figure supplement 2.** Structures of connectors 1, 2, and 3.

(*Figure 4D,E*). Thus, both connector 1 and connector 2 connect with PBS and PSII, suggesting that they may consolidate the anchor of PBS on PSII dimer.

Different from connectors 1 and 2, connector 3 mediates a connection between the lateral hexamer and the bridging PSII dimer C. The density map of connector 3 shows that connector 3 could be divided into one small part and one big part (*Figure 4—figure supplement 2D,E*). The small part is buried in the lateral hexamer's central cavity, and the big part extends out to connect with PSII. This structural feature is very similar with the structure of some rod linker proteins, such as $L_R6$ of *P. purpureum* PBS, with the N-terminal rigid domain occupying the central cavity of a hexamer and the C-terminal region extending out, suggesting that connector 3 might be the linker protein of the lateral hexamer. The structure indicates that the lateral hexamer together with connector 3 stabilizes PBS's connection with PSII in the interconnected array of the PBS–PSII supercomplex.

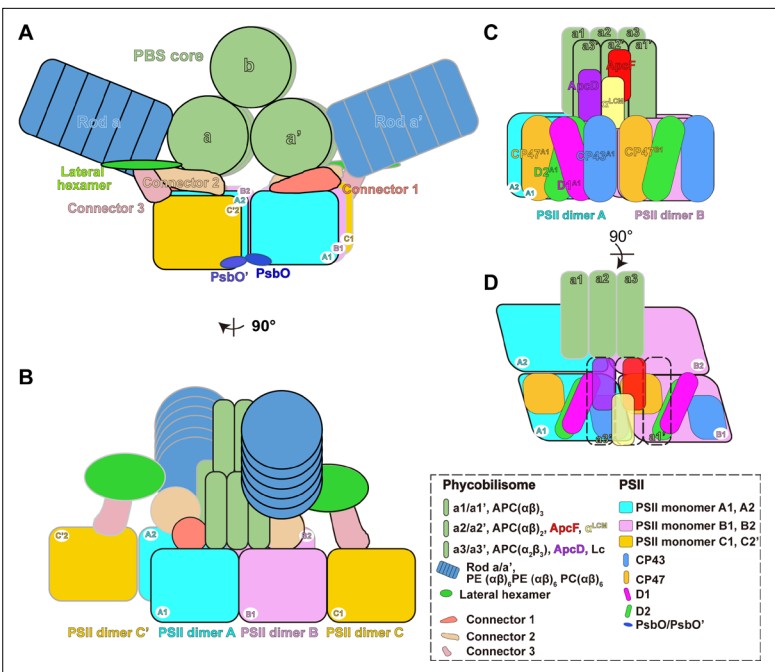

**Figure 5.** Schematic model of the PBS–PSII supercomplex. (**A**) The front view of the schematic model of the PBS–PSII supercomplex. For PBS, only core and Rod a are represented for clarity. In the PBS core, a and a' represent the two basal cylinders. Each cylinder consists of three APC trimers. b represents the top cylinder containing two APC trimers. (**B**) The side view of (**A**). Each PBS interacts with six PSII monomers: the two basal cylinders a and a' of the PBS core directly connect with PSII monomers A1, A2, B1, and B2. The connections are consolidated by connector 1 and connector 2; the lateral hexamers indirectly interact with the PSII monomers C1 and C'2 in the two peripheral PSII bridging dimers C and C' through connector 3. Two PsbO subunits from the adjacent PSII dimers bind with each other and may promote stabilization of the PBS–PSII supercomplex. The PBS–PSII supercomplexes regularly stack face-to-face into the interconnected arrays, forming the organized network on the thylakoid membrane. (**C**, **D**) The detailed connections between PBS and PSII. $\alpha^{LCM}$ interacts with CP43 and CP47 of PSII monomer, which may provide a way to funnel light energy to the reaction center. ApcD connects with CP43, suggesting another energy transfer way. (**D**) Highlights the connections of $\alpha^{LCM}$ - CP43$^{A1}$, $\alpha^{LCM}$–CP47$^{B1}$, and ApcD–CP43$^{A1}$. CP43$^{A1}$, CP43 of PSII dimer A1; CP47$^{B1}$, CP47 of PSII dimer B1.

## Discussion

### Organization of PBS and PSIIs

PBS is classified into different structural types, including hemidiscoidal (*Chang et al., 2015*), hemiellipsoidal (*Arteni et al., 2008*), block type (*Zhang et al., 2017*), bundle type (*Guglielmi et al., 1981*) and rod type (*Chen et al., 2009*). These results in diverse organizational structures between PBS and PSII. Hemidiscoidal PBSs of the *Synechocystis* strain PCC 6803 and the *Anabana* sp. strain PCC 7120 are associated with one PSII dimer (*Chang et al., 2015*; *Rast et al., 2019*). The rod-like AmPBS is composed of four hexamers and is adjacent to one PSII dimer (*Chen et al., 2009*; *Niedzwiedzki et al., 2019*). The hemiellipsoidal PBS connects with monomeric PSII at a ratio from 1:1 to 1:4 (*Arteni et al., 2008*; *Cunningham et al., 1989*; *Lange et al., 1990*; *Ohki et al., 1987*; *Takahashi et al., 2009*). Using in situ cryo-ET, our work reports the native PBS–PSII supercomplex structure of red algae. The structure reveals a different ratio (1:6) of hemiellipsoidal PBS and PSII on the native thylakoid membrane (*Figure 5A,B*). Each PBS directly interacts with two PSII dimers and indirectly interacts with two PSII monomers from two peripheral bridging PSII dimers, forming the interconnected array of the PBS–PSII supercomplex with the support of three unassigned connectors. With 14° rotation and 30 Å shift of PSII dimer pair relative to PBS core, the terminal emitters, ApcD and $\alpha^{LCM}$, in the two symmetric a2a3 and a2'a3' hexamers of the core, connect with CP43 of PSII monomers A1 and B2, and CP47 in PSII monomers B1 and A2 (*Figure 5*). A similar geometry of the PBS–PSII supercomplex was reported in cyanobacteria and red algae possessing hemidiscoidal PBS, suggesting that hemidiscoidal PBSs use the same strategy for energy transfer from PBS to PSII (*Arteni et al., 2009*; *Arteni et al., 2008*). This

structure also enables the extra lateral hexamers in *P. purpureum* to associate with PSII monomers C1 and C'2 through connector 3 (*Figure 5A,B*).

In this work, we focused on the PBS–PSII structures in the ordered distribution regions, where both PBS and PSII are neatly arranged according to a strict stoichiometry of 1:6. Previous research conducting fraction freezing analysis (*Lange et al., 1990*) has found that the PBS–PSII complexes on random distribution regions could have lower stoichiometry (between 1:1 and 1:4), which are represented as an PSII monomer, dimer, trimer, or double dimer. These observations show the possible diversity of PBS–PSII organization on the thylakoids.

## The connection between PBS and PSIIs

Extensive studies have shown that α$^{LCM}$ and ApcD are the terminal emitters of the energy coupling between PBS and PSII (*Zhao et al., 1992*; *Zlenko et al., 2019*). In our structures, we detailed the connections of α$^{LCM}$ and ApcD with PSII, which contribute to the energy flows from PBS to PSII (*Figure 3*). ApcF was also reported to be involved in the energy transfer in cyanobacteria (*Calzadilla et al., 2019*; *Chang et al., 2015*). However, we could not find a solid connection between ApcF and PSII. Thus, it is very likely that ApcF only plays an auxiliary role for direct energy transfer to reaction centers in red algae.

Additionally, these in situ structures reveal that three extra connectors are associated with PSII by the PBS core, Rod a, and the lateral hexamer, respectively. While the current resolution limited additional detailed analysis of the connectors, we attempted to estimate the molecular weights of these connectors at around 30–80 kDa according to the size and shape (*Figure 4—figure supplement 2*).

We found a class of PSII dimer C (PSII dimer C, C', and C" in *Figure 2A*) that does not directly bind to PBS but is closer to the lateral hexamer. Since PSII dimer C can only absorb energy from the independent lateral hexamer instead of the whole PBS through connector 3, they could have fewer bilins feeding into them than the other PSIIs and could become saturated with light at higher intensities. If

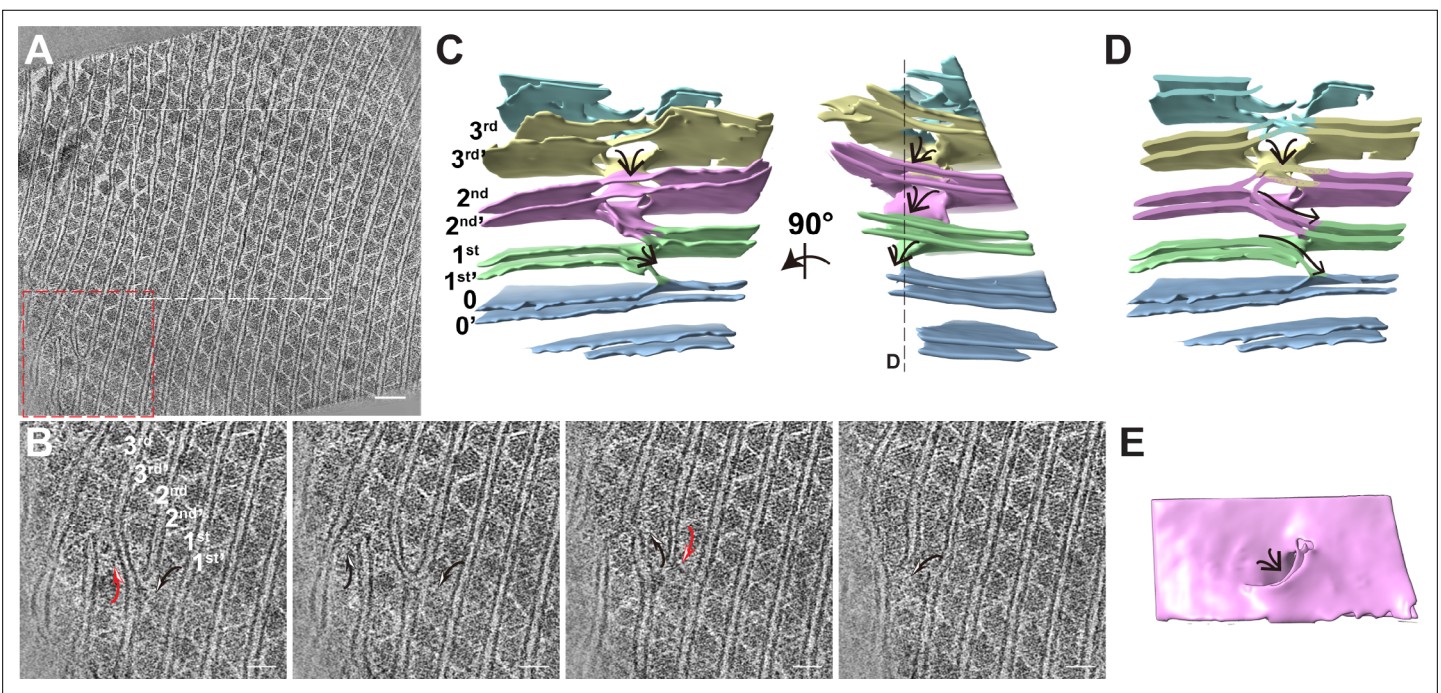

**Figure 6.** The thylakoid membrane builds perforations and stairs by branching and fusing with the neighboring membrane. (**A**) Typical tomogram slice. The white box indicates the tomogram region shown in *Figure 1A*; red box represents the membrane branching and fusing area. Scale bar, 100 nm. (**B**) Sequential slices back and forth through the representative tomogram in the red box area of (**A**), showing the membrane branches off one membrane and fuse with the neighboring membrane. The black arrow shows the fusion membrane; Red arrow shows the branching event. Scale bars, 50 nm. (**C**) 3D segmentation model of the red box area of (**A**) in the absence of the PBS–PSII supercomplexes to show the 3D morphology of the thylakoid. The two-line arrow indicates the membrane stairs. The numbers label different layers of membrane. (**D**) The clip view of the 3D segmentation model. The clip position is showed in the right panel of (**C**). The arrows show the membrane stairs. (**E**) The membrane top view shows the perforation on the top of the membrane stair. Figure accompanied by *Video 5*.

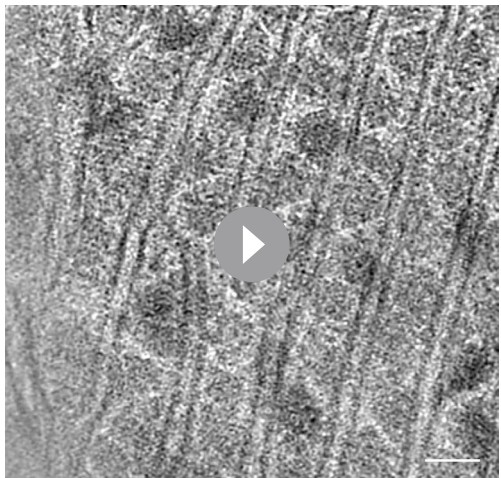

**Video 5.** Sequential slices back and forth through the representative tomogram slice to show thylakoid membrane branching and fusing areas. Related to Figure 6A–E. Scale bar, 20 nm.
https://elifesciences.org/articles/69635/figures#video5

the PSII dimer C is more sensitive to light intensity than the others, it could be involved in the transition between the ordered distribution region and the random distribution region. The transient absorption spectroscopic measurements and time-resolved fluorescence spectroscopy at low temperatures (77k or 4 k) could help confirm this hypothesis.

The energy transfer from PBS to PSII could follow multiple pathways (*Figure 5*). The two terminal emitters, $\alpha^{LCM}$ and ApcD, funnel light energy to PSII by distinct structural connections. The chromophore of $\alpha^{LCM}$ mediates energy transfer via interaction with CP43 and CP47. In contrast, the chromophore of ApcD is mediated by interactions with CP43. Since connector 2 connects with both the basal PC hexamer of Rod a and CP47 of PSII, we speculate that alternative routes of energy transfer bypassing the PBS core to PSII could exist: connector 2 and connector 3 could mediate the energy flow from the basal hexamer of Rod a to PSII and from lateral hexamer to PSII, respectively. Ueno et al. have reported light energy flows from CP43 to CP47 by energy transfer between the neighboring PSII monomers (*Ueno et al., 2017*). Our in situ structure reveals that $\alpha^{LCM}$ transfers energy to CP43 as well as the neighboring CP47, may suggesting that $\alpha^{LCM}$ provides the modulation activity of energy flow between CP43 and the neighboring CP47. However, it is currently unclear whether the connector proteins play a specific role in the energy transfer between PBS and PSII, and this topic requires further study.

## The perforations and stairs of the thylakoids

In several tomograms, we observed the thylakoid membrane branching and fusion regions (*Figure 6*, *Video 5*). Within the limited thickness of the tomogram, we observed a small area of thylakoid membranes present branching and fusion events (*Figure 6A*). The two parallel thylakoid membranes are split into two branches. One branch approaches and fuses with another branch from the neighboring thylakoid membrane (*Figure 6A,B*). The 3D segmentation model shows that the branching and fusing membranes built 'stairs' between two neighboring thylakoid membranes (*Figure 6C,D*). On the top of the 'stairs' is the large perforation on the thylakoids (*Figure 6E*). These structures are very similar to the thylakoid membrane architecture in the cyanobacteria *Prochlorococcus* (*Ting et al., 2007*), *Synechococcus* sp. PCC 7942 and *Microcoleus* sp. reported by *Nevo et al., 2007*. They also observed perforations as well as branching and fusion of the membranes, which resulted in a highly connected network to allow water-soluble and lipid-soluble molecules to diffuse through the entire membrane network (*Nevo et al., 2007*). Thus, it is reasonable to deduce that the perforations and stairs of the thylakoid membrane in the red alga might play the same role, although it needs to be further investigated.

## Materials and methods

**Key resources table**

| Reagent type (species) or resource | Designation | Source or reference | Identifiers | Additional information |
|---|---|---|---|---|
| Strain, strain background (*Porphyridium purpureum*) | Porphyridium purpureum UTEX 2757 | https://utex.org/products/utex-lb-2757 | UTEX Culture Collection of Algae 2,757 | |
| Gene (*Porphyridium purpureum*) | uniprot | https://www.uniprot.org/uniprot/ | UP000324585 | Proteome |

*Continued on next page*

*Continued*

| Reagent type (species) or resource | Designation | Source or reference | Identifiers | Additional information |
|---|---|---|---|---|
| Software, algorithm | SerialEM | https://bio3d.colorado.edu/SerialEM/ | Version 3.1.8 | Data collection |
| Software, algorithm | Chimera | https://www.cgl.ucsf.edu/chimera/ | Version 10.15 | Visualization |
| Software, algorithm | Relion | https://www3.mrc-lmb.cam.ac.uk/relion/ | Version 2.1 | Subtomogram averaging |
| Software, algorithm | IMOD | https://bio3d.colorado.edu/imod/index.html | Version 4.9 | Tomogram reconstruction |
| Software, algorithm | TOMO3D | https://sites.google.com/site/3dem/imageprocessing/tomo3d | Version 2.0 | Tomogram reconstruction |
| Software, algorithm | I3 | https://www.electrontomography.org/ | Version 0.9.3 | Subtomogram averaging |

## Cell culture and cryo-EM sample preparation

*Porphyridium purpureum* (from UTEX Culture Collection of Algae, UTEX 2757) cell was cultured in Bold 1NV:Erdshreiber (1:1) half-seawater medium, bubbled with sterilizing filtered air at 22 °C, under a light–dark periodic ratio of 16 hr:8 hr, with a white light flux of about 37 μmol photons•m$^{-2}$•s$^{-1}$. Then red algal cells were harvested at about 1.5–3.0 months by centrifuge at 6000 g at room temperature for 10 min under dark, and the pellet was washed and re-suspended with fresh half-seawater culture medium under dark.

We used holy-carbon copper grids (Quantifoil R1.2/1.3, 200 mesh) for the cryo-EM sample preparation. Cryo-EM grids were prepared with Leica EM GP (Leica Company) at 20 °C and 100 % humidity. To minimize the environmental affection (*Kaňa et al., 2014*), we first prepared the plunge freezing equipment. Then red algal cells are harvested and immediately vitrified in the Leica EM GP chamber without light. A drop of 4 μL culture medium was added to the glow-discharged grids. After being blotted, the grids were plunged into liquid ethane and stored in liquid nitrogen. We prepared four to six grids for each session. Each grid took less than 1 min. Moreover, we covered a layer of foil paper on the cell volume to protect them from the light.

## Cryo-FIB lamella preparation

With the modified workflow, as *Schaffer et al., 2017* reported, cryo-EM grid was first transferred to Helios NanoLab G3 (FEI) system. A layer of Au was sputtered to the surface of cryo-EM sample to increase the conductivity. A layer of protective organometallic platinum was then deposited on the top of the sample with the GIS system. The working distance was 10 mm, and the GIS temperature was set to 46 °C. Ga$^{2+}$ ion beam was used to milling the cells at a 5° stage tilt. The beam current for rough milling was 0.79 nA and gradually decreased to 40 pA. The lamella was finally polished to about 150 nm in thickness with the beam current of 24 pA.

## Cryo-ET data collection

The cryo-ET data were collected with Titan Kiros Microscopy (Thermo Fischer Scientific) operated at a voltage of 300 kV and equipped with a Cs corrector, GIF quantum energy filter, and K2 Summit direct electron detector (Gatan Inc). All tilt series were recorded from 60° to –60° with SerialEM software (*Mastronarde, 2005*). The recording state was at a nominal magnification of 33,000× in counting mode with a pixel size of 3.421 Å•pixel$^{-1}$. Each stack was exposed for 2.4 s with an exposure time of 0.3 s per frame and recorded as a movie of 8 frames, resulting in the total dose rate of approximately 1.927 electrons per Å$^2$ for each stack. The tomographic tilt series were recorded from −50° to +66° with an increment of 2° using the unidirectional strategy. On average, 6 frames were collected for each image resulting in a total dose between 100 e$^-$/Å$^2$ and 110 e$^-$/Å$^2$ per tilt series. GIF was set to a slit width of 20 eV. The defocus ranged from −2.8 μm to −5.5 μm. MotionCor2 program was used to correct the beam-induced motion (*Zheng et al., 2017*).

## Cryo-ET reconstruction and sub-tomograms averaging

Fifty-one tilt series were aligned with the patch-tracking method in IMOD software (*Kremer et al., 1996*). The tomograms were reconstructed with TOMO3D scripts (*Agulleiro and Fernandez, 2015*). The tomograms reconstructed with simultaneous iterative reconstruction technique were used to manually particle-picking. To get high-resolution structures, we performed sub-tomogram averaging. Five thousand and thirty-seven sub-tomograms were firstly picked and aligned with i3 software to generate an initial model (*Winkler et al., 2009*). Tomograms reconstructed with weighted-back projection were used for further sub-tomogram averaging analysis. Then 75,310 manually picked sub-tomograms were extracted with RELION software (*Bharat et al., 2015*). CTF correction was estimated with CTFFIND4 (*Rohou and Grigorieff, 2015*) implemented in RELION. Dose damage compensation was performed with the script provided in the RELION tutorial. The first round of auto-refinement was performed with the binning four sub-tomograms. For the reconstruction of the double PBS–PSII supercomplex, the coordinates of first-round auto-refinement were shifted and carried on another round of 3D classification with global search. The good classes were selected and performed auto-refinement with the binning two sub-tomograms. After refinement, a resolution map at 15.6 Å was achieved. To improve the resolution, we reconstructed the PBS–PSII supercomplex with the refined coordinates and Euler angles obtained in the first round of rough refinement. These sub-tomograms were performed 3D classification without alignment. The good classes were chosen to do another round of auto-refinement with local search at 1.8 degrees. Finally, the sub-tomograms of the PBS–PSII supercomplex without binning were refined to 14.3 Å and postprocessed to 13.2 Å. During the auto-refinement, the datasets were automatically divided into two halves by RELION. The resolution was estimated using the gold-standard Fourier shell correlation with 0.143 criteria by ResMap (*Scheres and Chen, 2012*).

## Atomic model fitting and analysis

To analyze the double PBS–PSII and PBS–PSII sub-tomogram averaging maps, the atomic model of PBS built with single particle analysis and crystal model of PSII (PDB code 4YUU) were docked into the sub-tomogram averaging maps with 'fit in map' command in UCSF Chimera (*Pettersen et al., 2004*). The cross-correlation coefficient results are listed in *Supplementary file 1*. The geometry of the PBS–PSII supercomplex was measured with the 'distance' and 'angles' command in UCSF Chimera. To analyze the arrangement of the PBS–PSII supercomplex in the thylakoid membrane, the double PBS–PSII and PBS–PSII sub-tomogram averaging maps were re-mapped back into the tomograms with the refined orientations and positions taking use of home-made scripts. The segmentation of the thylakoid membrane was performed with TomoSegMemTV (*Martinez-Sanchez et al., 2014*), Amira software (FEI Visualization Sciences Group), and UCSF ChimeraX (*Pettersen et al., 2021*). Segmentation of the subunits of the double PBS–PSII and PBS–PSII supercomplex was conducted in UCSF Chimera with the 'Segmentation panel'. The densities of the subunits were then extracted by subtracting other parts with the subtract function. Surface rendering was further colored with the color zone panel.

## Acknowledgements

This work was supported by the National Basic Research Program (2016YFA0501101 and 2017YFA0504600 to S-FS, 2016YFA0501102 and 2016YFA0501902 to XL), the National Natural Science Foundation of China (31670745 and 31861143048 to S-FS, 31722015 to XL, 32000848 to JM), Advanced Innovation Center for Structural Biology (to XL), Tsinghua-Peking Joint Center for Life Sciences (to XL). We thank the staff at the Tsinghua University Branch of the National Protein Science Facility (Beijing) for their technical support on the Cryo-EM and High-Performance Computation platforms. We thank XM Li for participating in facility supports. We thank JL Lei for data collection; J Liu and S Li for their recommendations on computation; YC Wang for recommendations on phylogenetic analysis; S K Cheppali, XC Qin, S Sun, and Y N Xiao for comments on the manuscript.

## Additional information

### Funding

| Funder | Grant reference number | Author |
|---|---|---|
| National Basic Research Program | 2016YFA0501101 | Sen-Fang Sui |
| National Basic Research Program | 2017YFA0504600 | Sen-Fang Sui |
| National Basic Research Program | 2016YFA0501102 | Xueming Li |
| National Basic Research Program | 2016YFA0501902 | Xueming Li |
| National Natural Science Foundation of China | 31670745 | Sen-Fang Sui |
| National Natural Science Foundation of China | 31861143048 | Sen-Fang Sui |
| National Natural Science Foundation of China | 31722015 | Xueming Li |
| National Natural Science Foundation of China | 32000848 | Jianfei Ma |
| Advanced Innovation Center for Structural Biology | | Xueming Li |
| Tsinghua-Peking Joint Center for Life Sciences | | Xueming Li |

The funders had no role in study design, data collection and interpretation, or the decision to submit the work for publication.

### Author contributions

Meijing Li, Conceptualization, Data curation, Formal analysis, Investigation, Methodology, Software, Validation, Visualization, Writing – original draft; Jianfei Ma, Conceptualization, Data curation, Formal analysis, Funding acquisition, Investigation, Methodology, Validation, Visualization, Writing – original draft; Xueming Li, Sen-Fang Sui, Formal analysis, Funding acquisition, Project administration, Supervision, Validation, Writing - review and editing, designed project, designed project

### Author ORCIDs

Meijing Li http://orcid.org/0000-0003-3931-7905
Jianfei Ma http://orcid.org/0000-0002-2016-2280
Sen-Fang Sui http://orcid.org/0000-0001-5253-2592

### Decision letter and Author response

Decision letter https://doi.org/10.7554/eLife.69635.sa1
Author response https://doi.org/10.7554/eLife.69635.sa2

---

## Additional files

### Supplementary files

• Supplementary file 1. Cross-correlation coefficient (CCC) of crystal structures or single particle analysis model and sub-tomogram averaging map.

• Transparent reporting form

### Data availability

The cryo-EM density maps have been deposited in the Electron Microscopy Data Bank under the accession number EMD-31241, EMD-31242, EMD-31243, EMD-31244, EMD-31245.

The following dataset was generated:

| Author(s) | Year | Dataset title | Dataset URL | Database and Identifier |
|---|---|---|---|---|
| Sui S-F, X M Li, M J Li, Ma JF | 2021 | Structure of the PBS-PSII supercomplex from red algae | https://www.ebi.ac.uk/pdbe/entry/emdb/EMD-31241 | Electron Microscopy Data Bank, EMD-31241 |
| Sui S-F, X M Li, M J Li, Ma JF | 2021 | Structure of the double phycobilisome-photosystem II supercomplex from red alga | https://www.ebi.ac.uk/pdbe/entry/emdb/EMD-31242 | Electron Microscopy Data Bank, EMD-31242 |
| Sui S-F, X M Li, M J Li, Ma JF | 2021 | Cryo-electron tomography of phycobilisome-photosystem II supercomplex on the thylakoid | https://www.ebi.ac.uk/pdbe/entry/emdb/EMD-31243 | Electron Microscopy Data Bank, EMD-31243 |
| Sui S-F, X M Li, M J Li, Ma JF | 2021 | Cryo-electron tomography of phycobilisome-photosystem II supercomplex on the thylakoid | https://www.ebi.ac.uk/pdbe/entry/emdb/EMD-31244 | Electron Microscopy Data Bank, EMD-31244 |
| Sui S-F, X M Li, M J Li, Ma JF | 2021 | Cryo-electron tomography of phycobilisome-photosystem II supercomplex on the thylakoid | https://www.ebi.ac.uk/pdbe/entry/emdb/EMD-31245 | Electron Microscopy Data Bank, EMD-31245 |

The following previously published datasets were used:

| Author(s) | Year | Dataset title | Dataset URL | Database and Identifier |
|---|---|---|---|---|
| Ma JF, You X, Sun S, Wang XX, Qin S, Sui S-F | 2020 | Structure of the phycobilisome from the red alga Porphyridium purpureum | https://www.rcsb.org/structure/6KGX | RCSB Protein Data Bank, 6KGX |
| Ago H, Adachi H, Umena Y, Tashiro T, Kawakami K, Kamiya N, Tian L, Han G, Kuang T, Liu Z, Wang F, Zou H, Enami I, Miyano M, Shen J-R | 2016 | Crystal structure of oxygen-evolving photosystem II from a red alga | https://www.rcsb.org/structure/4YUU | RCSB Protein Data Bank, 4YUU |

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
