## [Decision Letter]

**Acceptance summary:**

Interactions between reaction centres and phycobilisome light-harvesting complexes is crucial for photosynthesis in cyanobacteria and red algae, which between them account for a major fraction of photosynthesis on earth. This work uses in situ cryo-electron tomography to understand those interactions which were previously not amenable to study because they are too labile to survive isolation of the complexes.

**Decision letter after peer review:**

Thank you for submitting your article "In situcryo-ET structure of phycobilisome-photosystem II supercomplex from red alga" for consideration by *eLife*. Your article has been reviewed by 3 peer reviewers, and the evaluation has been overseen by a Reviewing Editor and Volker Dötsch as the Senior Editor. The following individual involved in review of your submission has agreed to reveal their identity: Conrad W Mullineaux (Reviewer #3).

Essential revisions:

1) The reviewers were overall excited by the insights your in-situ cryo-ET analysis gave into the organization of the PBS/PSII complex. However in some places we felt the descriptions of atom level interactions were not justified by the data. In other regions the descriptions are speculative but worth including.

Please remove the following discussions:

– Page 4 lines 35-36. The claim of a 45 deg rotation is unsupported by the data and should be removed.

– Page 7, lines 1-11. This section is too speculative and conclusions are drawn regarding missing residue of a loop which are really not supported by the data.

Please modify other parts of the manuscript to make it clear that the descriptions are more speculative:

– Page 7, lines 18-20.

– Page 8, lines 5-13.

– Page 8, lines 20-24.

2) The methods section needs more detail to allow readers to understand how the structure were solved. Please see the comments by reviewer 2 below.

*Reviewer #1:*

This manuscript reports the in-situ structure of the red algal phycobilisome while still bound to the thylakoid membrane. It is a follow-up to the same group's impressive cryo EM structure of the isolated phycobilisome from the same organism (Ma et al. Nature 2020). While the resolution of the current structure is significantly lower than that of the isolated complex, (13+ A vs 2.8 A), it has the advantage of revealing the organization in the membrane, the connections to Photosystem II and additional details of the structure that may be too labile to survive the isolation of the complex.

Overall, I think this is an impressive study. Of particular note is the discovery of the Type C PSII, which I don't think has been previously reported. The emphasis on the excitation energy coupling pathways to PSII is also important new information, although the relatively low resolution of the current structure makes that analysis less certain.

I suggest that the authors emphasize in clearer terms that this is an in-situ structure that is complementary to the structure of the isolated complex and reveals elements of the in vivo organization that can never be obtained through structures of the isolated complex, no matter how high the resolution. I think some readers may not appreciate that point sufficiently, and the current manuscript does not emphasize this critical point.

*Reviewer #2:*

Li et al., report the in situ structure of the phycobilisome (PBS) – photosystem II (PSII) complex in P. purpureum cells. They prepare cells by cryo-FIB/SEM, and collect cryo-tomograms around thylakoid membrane regions that are decorated by PBS/PSII regular arrays. They then perform subtomogram averaging of PBS pairs and single complexes to obtain structures at resolutions ranging between 13 and 16 A. These allow fitting of available atomic models and analysis of the molecular architecture and interfaces that characterise the complex.

The authors are able to confirm the overall architecture of PBS which was obtained from previous studies, but also reveal a number of novel features, including unexpected stoichiometry of PBS:PSII, and some uncharacterised interfaces. The results are organised into three main sections:

– Overall structure of the PBS-PSII supercomplex

Here the authors describe how each unit is composed of a PBS and three PSII dimers, two of which contact PBS directly, and two seem connected through some novel densities (here called 'lateral densities'), which are assigned to extra PBS hexamer. This was not reported before and it might be an interface that is stable only in an in situ context. This section is strong and the conclusions are well justified.

– Interaction pattern between PBS and PSII

Here the authors utilise the fitted atomic models to analyse interfaces between PBS and the PSII dimers at the molecular level. While the resolution of their subtomogram averaging map is sufficient for rigid body fitting of the relevant structures, I find the analysis here too speculative, particularly when it comes to analysis of subdomains (i.e. the PB-loop and the 33 missing residues, and the 117deg rotation of CP43 N-terminus).

The claim of two pathways for electron transfer and their description is unsupported by the data.

– Supplementary interactions intermediated by three connector proteins

Here the authors identify regions of density that cannot be explained with the fitted atomic models, and cautiously assign them to connecting proteins. They then analyse the positions of these connecting proteins and describe the interfaces they bridge. While the discovery of these new densities and the overall description of their contacts is convincing, I feel that again the authors draw conclusions at a level of molecular detail that is too speculative given the resolution.

Overall this is a novel and solid paper with regards to the overall description of PBS/PSII in situ, its stoichiometry and general architecture, which should be of interest to the fields of electron transport and in situ cryo-tomography. However some of the claims are not justified by the data, in particular those that analyse interactions in molecular detail.

Comments and recommendations below in no particular order

– While overall the paper is well written and logically explained, there are several typographical and grammatical errors. Moreover, certain sections are unclear and rephrasing could improve legibility significantly. For example: Page 4, Lines 19-23, Page 5, Lines 27-30, Page 6, Lines 22-32, Page 7 Line 16.

– It is not clear from the methods if the maps were post-processed in any way, and low-pass filtered at the stated resolution. They should be filtered so as to remove the noisy features visible in the figures.

– Red boxes in Figure supplement 1B,C: it would be useful to see what is included also along the z direction.

– Page 4 lines 35-36. The claim of a 45 deg rotation is unsupported by the data and should be removed.

– I found the nomenclature in Figure 2A a bit confusing, would it not be easier to call subunits A1,A2, B1,B2,C1,C2, A'1,A'2,[…],C'1,C'2?

– As far as I understand, Figurer 2 shows two different maps in panel A and B. This should be clearly stated in the legend.

– For general readers some introduction to PBS architecture is required: what are Rod a, Rod b, etc?

– Page 6, lines 28-32. It is unclear whether the authors claim or not that the extra hexamer is a PE hexamer, as there are some contradictory sentences.

– There are a series of unsupported claims:

– Page 7, lines 1-11. This section is too speculative and conclusions are drawn regarding missing residue of a loop which are really not supported by the data.

– Page 7, lines 18-20. This sentence is not supported by the data. There is also no description in the methods of how the 117 deg angle was derived (flexible fitting?).

– Page 8, lines 5-13. As above

– Page 8, lines 20-24. As above

I suggest the authors limit themselves to a general description of which domain interact with each other rather than going into sub-domain or even residue-level considerations. There is value in their description of the architecture and interfaces, if only it would be made in less detailed terms.

– Page 7, lines 24-28. The basis for the claim are unclear.

– Re: connector proteins, how were they identified? By difference mapping with the fitted atomic model? This needs to be stated clearly.

– Many details are missing in the methods (at least those parts I can understand: tomogram collection and subtomogram abveraging). What was the total dose across the tilt series (or tilt increment)? There is no reference at all to ctf: was it corrected (I assume so within relion)? How was it estimated? Was dose damage compensation performed? Was the dataset divided in two halves according to the gold standard? In consequence, are the FSC reported gold standard FSCs?

– The authors need to add the FSC (and possibly local resolution estimations) as supplementary figures.

*Reviewer #3:*

The authors were able to convincingly identify a phycobilisome-photosystem II complex in their cryo-electron tomograms of chloroplasts from a red alga, and to obtain high-resolution structural models by sub-tomogram averaging and fitting parts of known structures to the tomographic data. A crucial feature of work is that it reports structures in situ, in intact flash-frozen sections. That's important for 2 reasons. Firstly, the in situ data reveals the larger-scale organisation of the complexes in the membrane, which is crucially important for photosynthetic performance. Secondly, the association between phycobilisomes and reaction centres appears rather plastic and is highly unstable (at least in vitro), raising the possibility of significant artefacts in any preparation of isolated complexes. So, in situ information of this sort is really needed to provide convincing answers. I am not well qualified to judge the technical aspects of the tomographic analysis, but it all looks sound as far as I am able to judge. The manuscript is very well-written and well-presented, and there is an excellent, balanced discussion of the findings.

---

## [Author Response]

Essential revisions:1) The reviewers were overall excited by the insights your in-situ cryo-ET analysis gave into the organization of the PBS/PSII complex. However in some places we felt the descriptions of atom level interactions were not justified by the data. In other regions the descriptions are speculative but worth including.Please remove the following discussions:– Page 4 lines 35-36. The claim of a 45 deg rotation is unsupported by the data and should be removed.– Page 7, lines 1-11. This section is too speculative and conclusions are drawn regarding missing residue of a loop which are really not supported by the data.Please modify other parts of the manuscript to make it clear that the descriptions are more speculative:– Page 7, lines 18-20.– Page 8, lines 5-13.– Page 8, lines 20-24.2) The methods section needs more detail to allow readers to understand how the structure were solved. Please see the comments by reviewer 2 below.

Thank you for your precious comments and summary. We have deleted the discussions not supported by the data (P. 4 lines 35-36; P. 7 lines 1-11) in the revised manuscript. According to reviewers ' constructive suggestions, we have also rewritten the speculative descriptions, which have been explained in the point-by-point responses to reviewers' comments. In the revised manuscript, all the experimental details suggested by reviewers have been included.

Reviewer #1:This manuscript reports the in-situ structure of the red algal phycobilisome while still bound to the thylakoid membrane. It is a follow-up to the same group's impressive cryo EM structure of the isolated phycobilisome from the same organism (Ma et al. Nature 2020). While the resolution of the current structure is significantly lower than that of the isolated complex, (13+ A vs 2.8 A), it has the advantage of revealing the organization in the membrane, the connections to Photosystem II and additional details of the structure that may be too labile to survive the isolation of the complex.Overall, I think this is an impressive study. Of particular note is the discovery of the Type C PSII, which I don't think has been previously reported. The emphasis on the excitation energy coupling pathways to PSII is also important new information, although the relatively low resolution of the current structure makes that analysis less certain.I suggest that the authors emphasize in clearer terms that this is an in-situ structure that is complementary to the structure of the isolated complex and reveals elements of the in vivo organization that can never be obtained through structures of the isolated complex, no matter how high the resolution. I think some readers may not appreciate that point sufficiently, and the current manuscript does not emphasize this critical point.

We appreciate your insightful suggestions. We have incorporated your suggestions into the last paragraph of the introduction part (Page 4, lines 9-12, lines 16-17).

Reviewer #2:Li et al., report the in situ structure of the phycobilisome (PBS) – photosystem II (PSII) complex in P. purpureum cells. They prepare cells by cryo-FIB/SEM, and collect cryo-tomograms around thylakoid membrane regions that are decorated by PBS/PSII regular arrays. They then perform subtomogram averaging of PBS pairs and single complexes to obtain structures at resolutions ranging between 13 and 16 A. These allow fitting of available atomic models and analysis of the molecular architecture and interfaces that characterise the complex.The authors are able to confirm the overall architecture of PBS which was obtained from previous studies, but also reveal a number of novel features, including unexpected stoichiometry of PBS:PSII, and some uncharacterised interfaces. The results are organised into three main sections:– Overall structure of the PBS-PSII supercomplexHere the authors describe how each unit is composed of a PBS and three PSII dimers, two of which contact PBS directly, and two seem connected through some novel densities (here called 'lateral densities'), which are assigned to extra PBS hexamer. This was not reported before and it might be an interface that is stable only in an in situ context. This section is strong and the conclusions are well justified.– Interaction pattern between PBS and PSIIHere the authors utilise the fitted atomic models to analyse interfaces between PBS and the PSII dimers at the molecular level. While the resolution of their subtomogram averaging map is sufficient for rigid body fitting of the relevant structures, I find the analysis here too speculative, particularly when it comes to analysis of subdomains (i.e. the PB-loop and the 33 missing residues, and the 117deg rotation of CP43 N-terminus).The claim of two pathways for electron transfer and their description is unsupported by the data.– Supplementary interactions intermediated by three connector proteinsHere the authors identify regions of density that cannot be explained with the fitted atomic models, and cautiously assign them to connecting proteins. They then analyse the positions of these connecting proteins and describe the interfaces they bridge. While the discovery of these new densities and the overall description of their contacts is convincing, I feel that again the authors draw conclusions at a level of molecular detail that is too speculative given the resolution.Overall this is a novel and solid paper with regards to the overall description of PBS/PSII in situ, its stoichiometry and general architecture, which should be of interest to the fields of electron transport and in situ cryo-tomography. However some of the claims are not justified by the data, in particular those that analyse interactions in molecular detail.

Thank you for your precious comments. We have carefully considered your valuable

suggestions and comments. We have dealt with your precious suggestions and responded to each comment. As a summary, we have re-analyzed the interaction interfaces between PBS and PSII at the domain level in the revised manuscript with the new figures and videos, as suggested by you. The interfaces provide convincing evidence for the multiple energy transfer.

We have clarified how we found the new densities and discussed their connections with PBS and PSII at a domain level. The claims that are not justified by the data have been removed.

Comments and recommendations below in no particular order– While overall the paper is well written and logically explained, there are several typographical and grammatical errors. Moreover, certain sections are unclear and rephrasing could improve legibility significantly. For example: Page 4, Lines 19-23, Page 5, Lines 27-30, Page 6, Lines 22-32, Page 7 Line 16.

We appreciate your insightful suggestions. We have rewritten these unclear descriptions.

– It is not clear from the methods if the maps were post-processed in any way, and low-pass filtered at the stated resolution. They should be filtered so as to remove the noisy features visible in the figures.

Thanks for noticing this. The previous PBS-PSII supercomplex map at a resolution of 13.2 Å is post-processed with the program implemented in Relion software. To evade the noisy features, we have updated all the figures with the non-post-processing map at 14.3 Å resolution, which does not present noisy features. We have provided a new Figure 2—figure supplement 1 to show the resolution estimation and local resolution maps for the double PBS-PSII supercomplex and PBS-PSII supercomplex.

– Red boxes in Figure supplement 1B,C: it would be useful to see what is included also along the z direction.

We appreciate your insightful suggestions. We have updated Figure 1—figure supplement 1B, 1C with the Z-direction project. Moreover, we have provided two related movies (Video 2 corresponding to panel B and Video 3 corresponding to panel C) to show the sequential slices back and forth in Z-direction through the tomographic volume. To provide a clear three dimensional view of the tomographic volumes we used in the manuscript, we provided Video 1 (Corresponding to Figure 1A), Video 5 (Corresponding to Figure 6B).

– Page 4 lines 35-36. The claim of a 45 deg rotation is unsupported by the data and should be removed.

We appreciate your insightful suggestions. We have removed the discussion.

– I found the nomenclature in Figure 2A a bit confusing, would it not be easier to call subunits A1,A2, B1,B2,C1,C2, A'1,A'2,[…],C'1,C'2?

We appreciate your insightful suggestions. We have replaced the annotations throughout the revised manuscript and the corresponding figures.

– As far as I understand, Figurer 2 shows two different maps in panel A and B. This should be clearly stated in the legend.

Thank you for noting this. We have rewritten the legend as follows:

“(A) The density map of double PBS-PSII supercomplex at a resolution of 15.6 Å, presented in two perpendicular views. The center-to-center distance of two adjacent PBSs is approximately 345 Å. Two random circles indicated by black or red dashed lines mark the six PSII monomers binding with each of the two PBSs. PBS1 is associated with PSII monomer A1, A2, B1, B2, C1, and C'2. PBS2 connects with A'1, A'2, B'1, B'2, C'1, C’’2. The surface threshold is 0.09. (B) The density map of the PBS-PSII supercomplex at a resolution of 14.3 Å fitted with the single particle model of PBS (EMDB code EMD-9976, PDB code 6KGX) and X-ray structure of PSII (PDB code 4YUU), presented in two perpendicular views. The lateral hexamer was fitted with the single-particle model of the Rod a distal PE hexamer. The surface threshold is 0.059. (C) The magnified image shows that the two PsbO subunits bind with each other at the interface of the adjacent PSII dimers (Inset). A map of PSII dimers A and B, segmented from (B) with the same surface threshold level. The arrow indicates the binding site.”

– For general readers some introduction to PBS architecture is required: what are Rod a, Rod b, etc?

We appreciate your insightful suggestions. We have included the introduction about Rod a, Rod b, PBS core, PSII composition in the introduction section as follows:

“The cryo-EM structure of PBS in red alga Porphyridium purpureum (Ma et al., 2020) shows that the PBS consists of a tricylindrical PBS core,14 rods (Rod a – Rod g, Rod a’– Rod g’), 8 individual extra PE (αβ)_6_ hexamers (Ha – Hd, Ha’– Hd’) and 24 individual extra PE α or β subunits. The core contains one top cylinder B, composed of two allophycocyanin (APC) (αβ)_3_ trimers, and two bottom cylinders (A and A’), each of which is assembled by three APC trimers (A1-A3, A2 and A3 form APC hexamer). The rods consist of phycoerythrin (PE) and phycocyanin (PC) hexamers or only PE hexamers. For example, Rod a is composited of one basal PC hexamer and two distal PE hexamers. Thus, the energy absorbed by Rod transfers uni-directionally from the distal PE to the basal PC, and then funnels to APC in the core, and eventually to the two terminal emitters, including chromophores in the core–membrane linker protein (L_CM_, also called ApcE) (Capuano et al., 1991; Lundell et al., 1981; Tang et al., 2015) and allophycocyanin D (ApcD) (Glazer and Bryant, 1975; Peng et al., 2014).”

(Page 2, lines 32 - 36, Page 3, lines 1-7).

"The main subunits of PSII include the reaction center D1, D2 protein, and chlorophyll-a binding proteins, CP43 and CP47. Both CP43 and CP47 subunits have been reported to mediate the energy transfer from PBS to the reaction center (Ueno et al., 2017)."

(Page 3, lines 16-19).

– Page 6, lines 28-32. It is unclear whether the authors claim or not that the extra hexamer is a PE hexamer, as there are some contradictory sentences.

We appreciate your insightful suggestions. We have rewritten the unclear description. In the revised manuscript, we claimed the lateral hexamer might be a PE hexamer as follows:

“To analyze the lateral densities on both sides of PBS connecting PBS to PSII dimer C, we

further improved the resolution of the PBS-PSII supercomplex to 13.2 Å after post-processing from the 14.3 Å resolution map (Figure 2—figure supplement 1D). The slice view of the PBSPSII map shows that the densities are round discs with stronger densities in the center (Figure 2—figure supplement 3E–G). These features are reminiscent of the structure of the extra hexamer found in the single-particle cryo-EM structure of P. *purpureum* PBS. Moreover, the lateral densities are of the same size and shape as the extra hexamer. As all eight extra hexamers in P. *purpureum* PBS are PE hexamers, we hypothesize that the lateral density could be contributed by a PE hexamer, which is referred to as the lateral hexamer (Figure 2B). Further analysis indicated that the lateral hexamer connects with the bottom hexamer of Rod a and the second hexamer of Rod e, which are the PC and PE hexamers, respectively (Figure 2—figure supplement 3E).” (Page 6, lines 34 -36, Page 7, lines 1-8).

– There are a series of unsupported claims:– Page 7, lines 1-11. This section is too speculative and conclusions are drawn regarding missing residue of a loop which are really not supported by the data.– Page 7, lines 18-20. This sentence is not supported by the data. There is also no description in the methods of how the 117 deg angle was derived (flexible fitting?).– Page 8, lines 5-13. As above– Page 8, lines 20-24. As aboveI suggest the authors limit themselves to a general description of which domain interact with each other rather than going into sub-domain or even residue-level considerations. There is value in their description of the architecture and interfaces, if only it would be made in less detailed terms.

Thank you for noting this. We have removed the inappropriate descriptions and re-analyzed the interaction at the domain level by rephrasing the whole section 'Interaction pattern between PBS and PSII' (Page 7, lines 12-28) with the new figures (Figure 3, Figure 3—figure supplement 1). We do not paste the text here.

In addition, we have rewritten the unclear description of Page 8, lines 5-13, and lines 20-24. The revised parts are rephrased on Page 8, lines 6-23 with new figures (Figure 4). We do not paste the text here.

– Page 7, lines 24-28. The basis for the claim are unclear.

Thank you for noting this. We think that the claim is too speculative and we deleted this

claim. According to the reviewer’s suggestion, we have rephrased the whole section of

“Interaction pattern between PBS and PSII” with a general description of subunit interaction. (Page 7, lines 12-28)

– Re: connector proteins, how were they identified? By difference mapping with the fitted atomic model? This needs to be stated clearly.

We appreciate your insightful suggestions. Yes, we identified the connector proteins by

difference mapping with the fitted atomic models. We have rewritten the unclear description as follows:

“After docking the atomic models of PBS, PSII dimer and two lateral hexamers into the density map of PBS-PSII supercomplex, we still observed three extra densities that are not occupied by any model. Two of them are associated with the PBS and PSII (Figure 4—figure supplement 1 A), and the third one is associated with the lateral hexamer and PSII (Figure 4—figure supplement 1B). Since we could not identify the proteins corresponding to these densities, we temporarily deemed them "connectors" (connector 1 – 3; Figure 4, Figure 4—figure supplement 2A–E).” (Page 7, lines 32-35)

– Many details are missing in the methods (at least those parts I can understand: tomogram collection and subtomogram abveraging). What was the total dose across the tilt series (or tilt increment)? There is no reference at all to ctf: was it corrected (I assume so within relion)? How was it estimated? Was dose damage compensation performed? Was the dataset divided in two halves according to the gold standard? In consequence, are the FSC reported gold standard FSCs?

We appreciate your insightful suggestions. We have included the missed details in the

method section as follows.

The tomographic tilt series were recorded from -50° to +66° with an increment of 2° using the unidirectional strategy. (Page 13, lines 23-24)

CTF correction was estimated with CTFFIND4 (Rohou and Grigorieff, 2015) implemented in RELION. Dose damage compensation was performed with the script provided in the RELION tutorial. (Page 13, lines 38-39, Page 14, line 1)

During the auto-refinement, the datasets were automatically divided into two halves by RELION. (Page 14, lines 11-12)

– The authors need to add the FSC (and possibly local resolution estimations) as supplementary figures.

We appreciate your insightful suggestions. We have provided a new supplementary figure, Figure 2—figure supplement 1.